# Enhancement in Performance and Reliability of Fully Transparent a-IGZO Top-Gate Thin-Film Transistors by a Two-Step Annealing Treatment

**DOI:** 10.3390/nano15060460

**Published:** 2025-03-19

**Authors:** Shuaiying Zheng, Chengyuan Wang, Shaocong Lv, Liwei Dong, Zhijun Li, Qian Xin, Aimin Song, Jiawei Zhang, Yuxiang Li

**Affiliations:** 1Shandong Technology Center of Nanodevices and Integration, and School of Integrated Circuits, Shandong University, Jinan 250101, China; 201912230@mail.sdu.edu.cn (S.Z.); wangchengyuan@sdu.edu.cn (C.W.); shaoconglv@mail.sdu.edu.cn (S.L.); dlw00001997@163.com (L.D.); lizhijun@hlmc.cn (Z.L.); xinq@sdu.edu.cn (Q.X.); a.song@manchester.ac.uk (A.S.); 2State Key Laboratory of Crystal Materials, Shandong University, Jinan 250100, China; 3School of Electrical and Electronic Engineering, University of Manchester, Manchester M13 9PL, UK; 4Institute of Nanoscience and Applications, Southern University of Science and Technology, Shenzhen 518055, China

**Keywords:** amorphous InGaZnO_4_, top-gate thin-film transistor, two-step annealing treatment

## Abstract

A two-step annealing treatment was applied on a fully transparent amorphous InGaZnO4 (a-IGZO) top-gate thin-film transistor (TG-TFT) to improve the device performance. The electrical properties and stabilities of a-IGZO TG TFTs were significantly improved as the first-annealing temperature increased from 150 °C to 350 °C with a 300 °C second-annealing treatment. The a-IGZO TG-TFT with the 300 °C first-annealing treatment demonstrated the overall best performance, which has a mobility of 13.05 cm^2^/(V·s), a threshold voltage (*V*_th_) of 0.33 V, a subthreshold swing of 130 mV/dec, and a *I*_on_/*I*_off_ of 1.73 × 10^8^. The *V*_th_ deviation (Δ*V*_th_) was −0.032 V and −0.044 V, respectively, after a 7200 s positive and negative bias stress under the gate bias voltage *V*_G_ = ±3 V and *V*_D_ = 0.1 V. The Photoluminescence spectra results revealed that the distribution and the density of defects in a-IGZO films were changed after the first-annealing treatment, whereas the X-ray photoelectron spectroscopy results displayed that contents of the oxygen vacancy and Ga-O bond varied in annealed a-IGZO films. In addition, a-IGZO TG-TFTs had achieved a transmittance of over 90%. Research on the effects of the first-annealing treatment will contribute to the fabrication of highly stable top-gate TFTs in the fields of transparent flexible electronics.

## 1. Introduction

The top-gate (TG)-structured amorphous InGaZnO_4_ (a-IGZO) thin-film transistor (TFT) was widely used in fields of fingerprint sensors, transparent displays, and flexible electronics [1,2,3,4,5,6] due to its high gate control capability and excellent stability. The high stability of a-IGZO TG-TFTs requires a certain special treatment on the a-IGZO active layer, such as annealing in different gas atmospheres, plasma gas treatment, or covering a passivation layer [5,7,8,9,10,11,12]. The aforementioned various treatments could improve the bonding state of metal ions and oxygen atoms both on the surface and interface, and inside the a-IGZO active layer [9,13].

For TG-TFTs, the gate dielectric layer is deposited after the semiconductor active layer. The bonding state in the a-IGZO active layer would degrade during the deposition of Al_2_O_3_ gate dielectric material in the fabrication of TG-TFTs [5,14,15,16]. Especially, the quantity of oxygen vacancy (*V*_O_) and the distribution of O defects in a-IGZO were significantly altered when the Al_2_O_3_ top-gate dielectric is grown through the atomic layer deposition (ALD) process, leading to the degeneration in the electrical performance of a-IGZO TG-TFTs [8,9,10,14,17]. The variation in oxygen-related states mainly originated from oxygen outgassing from a-IGZO during the deposition of Al_2_O_3_ in the ALD high-temperature vacuum environment [18]. Thermal annealing treatments in air or O_2_ after gate dielectric deposition have been considered to be an effective method to improve oxygen-related states at the a-IGZO-Al_2_O_3_ interface and within the a-IGZO and Al_2_O_3_ layers, and increase the film density and uniformity [5,14,15,16], which mainly relies on oxygen diffusion mechanisms [5,14,15,19]. However, the diffusion of external oxygen into the a-IGZO active layer is significantly hindered due to the passivation effect of the Al_2_O_3_ top-gate dielectric, resulting in only partly eliminating the degradation of a-IGZO. In addition, oxygen vacancies are passivated by the hydrogen diffusing from the precursor residue of trimethylaluminum (TMA) in Al_2_O_3_ and the deficiency of oxygen in a-IGZO lead to the negative degradation of the threshold voltage (*V*_th_) of a-IGZO TG-TFTs [16,19]. Therefore, another annealing treatment on a-IGZO in air or O_2_ before the gate dielectric deposition was proposed in some reports to resist the degradation of a-IGZO due to the formation of metal–oxygen (M-O) bonds, reducing the *V*_O_ and O defects and densifying a-IGZO films [20,21]. In previous reports, the effect on the performance of devices and the mechanism of the annealing treatment after gate dielectric deposition have been thoroughly investigated [5,11,16,22], while the effect and underlying mechanism of the annealing treatment of a-IGZO before the gate dielectric deposition are seldom reported.

In this work, a two-step annealing treatment was applied during the fabrication of the TG-TFT. The annealing treatment on the a-IGZO active layer before ALD-Al_2_O_3_ gate dielectric deposition is defined as the first annealing (first-annealing), while the annealing of a-IGZO TG-TFTs after Al_2_O_3_ coverage is defined as the second annealing (second-annealing). The mechanism on a-IGZO and a-IGZO TFTs of the second-annealing treatment had been reported in our previous study [16]. Herein, the effect and mechanism of the first-annealing treatment on the a-IGZO active layer and the stability of a-IGZO TG-TFTs were systematically investigated based on a fully transparent a-IGZO TG-TFT fabricated under different first-annealing conditions. The UV–Vis optical transmission spectrum was used to obtain the transmittance of the a-IGZO TG-TFT. The photoluminescence (PL) spectroscopy and X-ray photoelectron spectroscopy (XPS) were employed to analyze the first-annealing mechanism through investigating the distribution and the content of defects within the a-IGZO layer.

## 2. Materials and Methods

Figure 1a shows a schematic of the fabricating process sequence and major processing steps, respectively, of the top-gate bottom-contact a-IGZO TFT in this work. The fully transparent staggered TG-TFTs were fabricated on a cleaned SiO_2_ (100 nm)/Si substrate. Firstly, ITO source and drain (S/D) electrodes were deposited by radio-frequency (RF) magnetron sputtering (Denton Vacuum, Moorestown, NJ, USA) at 60 W in an Ar atmosphere under a pressure of 0.43 mtorr for 840 s. The S/D regions were defined by photolithography, and outside regions were etched by CH_4_ and H_2_ at 60 °C using inductively coupled plasma reactive-ion etching (ICP-RIE) (Oxford Instruments, London, UK). Secondly, the a-IGZO channel was deposited through RF magnetron sputtering at 90 W in a pure Ar atmosphere from a ceramic target with a molar ratio of In_2_O_3_:Ga_2_O_3_:ZnO = 1:1:1 at room temperature (RT). Right after the deposition of a-IGZO layer, the first-annealing treatment was carried out in air (the air humidity is 40%) for 1 h, which the first-annealing treatment temperatures were controlled between 150 °C and 350 °C with a step of 50 °C. The a-IGZO channel regions were defined as 10 μm × 10 μm by photolithography, and were etched by CH_4_ and H_2_ at 20 °C using ICP-RIE technology. Subsequently, the Al_2_O_3_ gate insulator (GI) was deposited through thermal-ALD (Kemicro, Jiaxing, China) at 150 °C with trimethylaluminum (TMA) and H_2_O as the aluminum and oxygen precursors, respectively, and N_2_ as the purge gas. The pulse times for TMA, H_2_O, and N_2_ were 0.03 s, 0.02 s, and 30 s, respectively. Finally, ITO gate electrodes were deposited by RF magnetron sputtering in an Ar atmosphere and defined by lift-off processes. The second-annealing treatment was carried out at 300 °C in air for 1 h after the fabrication of a-IGZO TFTs finished. Additionally, the layer thicknesses of the ITO S/D electrodes, a-IGZO channel, Al_2_O_3_ GI, and ITO gate electrode were measured using atomic force microscope (AFM) (Being Nano-Instruments, Guangzhou, China) and found to be 50 nm, 24 nm, 10 nm, and 50 nm, respectively.

All electrical measurements of the a-IGZO TG-TFTs were performed using an Agilent B1500A parameter analyzer connected to Cascade (CASCADE MICROTECH, Hsinchu City, Taiwan) EPS1500TRIAX (Keysight, Santa Rosa, CA, USA) probe station at RT in a dark ambient atmosphere. The PL spectrum was obtained using a Renishaw inVia Raman (Renishaw, London, UK) spectrometer equipped with a 325 nm wavelength laser. XPS spectra were obtained on a Thermo Scientific K-Alpha XPS (Thermo Scientific, Waltham, MA, USA) system with monochromatic Al Kα X-ray (1486.7 eV) radiation and were calibrated by the C 1s peak (~284.8 eV).

## 3. Results

### 3.1. Optical and Electric Characteristics of TG-TFTs

The transmittance of the fully transparent staggered a-IGZO TG-TFT was shown in Figure 1b. The transmittance of the a-IGZO TG-TFT obtained by the UV–Vis optical transmission spectra had achieved over 90% (with quartz as the substrate) in the visible light range. The bandgap width of the a-IGZO film was also calculated, as shown in the insert of Figure 1b. The bandgap of the fresh a-IGZO film (without annealing) was 3.56 eV, while the bandgap of the a-IGZO film with annealing at 300 °C for 1h was 3.58 eV. This might be due to the significant reduction in tail states in the a-IGZO film during annealing, which results in a blue-shifted optical bandgap [23]. The pleasant transmittance of the TG-TFT was mainly attributed to the wide bandgap of the a-IGZO material, which is benefit for applications in transparent flexible electronics and smart wearable devices.

Figure 2a,b displayed the transfer characteristics of TG-TFTs with different first-annealing treatments. As shown in Figure 2a,b, the TG-TFT without the first-annealing treatment did not show the switching characteristic, which indicated that the degradation of the a-IGZO layer without first-annealing was most severe after the deposition of Al_2_O_3_. During the deposition process of Al_2_O_3_, the a-IGZO film was exposed to a high-temperature vacuum environment for a long time, which led to the following oxygen evolution reaction in the a-IGZO film, M-O + M-O → M-M + O_2_↑, resulting in the loss of the O atom and the generation of oxygen vacancies in the a-IGZO film [18]. The first-annealing treatment could increase the oxygen content and M-O binding energy inside IGZO before the coverage of Al_2_O_3_, and then helping the a-IGZO film resist O deficiency caused by the Al_2_O_3_ deposition process. There was a significant change in the transfer characteristics of the transistor after the introduction of the first-annealing treatment. The TFTs with first-annealing at 150 °C and 350 °C did not exhibit obvious switching characteristics, which were attributed to the formation of extensive shallow donor states and the creation of new trap states in a-IGZO [9], respectively. The shallow donor states generated during the first-annealing treatment would release electrons into the conduction band of a-IGZO, and the increase in the carrier concentration led to the increase in the drain current, resulting in the improvement of the field-effect mobility. The TFTs with first-annealing treatment at 200 °C–300 °C exhibited good switching characteristics, with performance varying depending on the first-annealing temperature. The electrical parameters of the TG-TFTs, such as the threshold voltage (*V*_th_), subthreshold swing (*SS*), field-effect mobility (*μ*_sat_), and on/off current ratio (*I*_on_/*I*_off_), were all extracted from the saturated transfer curve, as summarized in Table 1. The *V*_th-sat_ was shifting positively with increasing first-annealing temperature up to 250 °C, and then switching to a negative shift with further increasing first-annealing temperature. The turn-around *V*_th_ shift was associated with changes in the quantity and distribution of *V*_O_ and O defects, and the bonding energy between metal ions and oxygen atom within the internal composition of a-IGZO films [9,17,24].

From Figure 2b and Table 1, the values of *SS* were 203 mV (200 °C), 137 mV (250 °C), and 130 mV (300 °C), and showing a declining trend as increasing the first-annealing temperature. It was reported that the *SS* was mainly influenced by the interfacial quality and defect states between the Al_2_O_3_ and the a-IGZO channel [25]. The excellent interfaces between a-IGZO and Al_2_O_3_ had been proven by the negligible hysteresis of the transfer curves as shown in Figure 2a,b. And the density of the trapping states (*N_t_*) at the interface was extracted from the *SS* by the following formula:(1)$$SS=\frac{\kappa_{B}T\;\ln\;10}{q}\left[1+\frac{q^{2}}{C_{ox}}N_{t}\right]$$
where *κ*_B_ is Boltzmann’s constant, *C*_ox_ is the unit area capacitance of 10 nm Al_2_O_3_, *T* was the temperature in Kelvin, and *q* is the electron charge. In this work, the *C*_ox_ was 408 nF/cm^2^ obtained from the capacitance–voltage measurement. The calculated *N_t_* value was within the range of 2.98 × 10^12^–6.10 × 10^12^ eV^−1^cm^−2^ as shown in Table 1, and this result was also agreement with the previously reported value 3.0 × 10^12^ eV^−1^cm^−2^ [25]. The improvement in *SS* performance with increasing the first-annealing temperature might be mainly attributed to the reduction of O defects and *V*_O_ in the a-IGZO film [26]. 

### 3.2. Electrical Stability Test on a-IGZO TG-TFTs

The electrical stability of a-IGZO TG-TFTs was further investigated. As depicted in Figure 3 and Figure 4, positive bias stress (PBS) and negative bias stress (NBS) measurements were performed on TG-TFTs under the test conditions of *V*_stress_ (gate bias voltage) = ±3 V at *V*_D_ = 0.1 V. As displayed in Figure 3a–c, transfer curves of all TG-TFTs shifted very slightly in the negative direction after a positive stress duration of 7200 s, demonstrating an excellent resistance to PBS. And this extraordinary stability was mainly due to the good interface between the a-IGZO channel and the Al_2_O_3_ insulator [27,28,29]. Moreover, the negative shift in transfer curves was likely caused by the H-doping effect, which was resulting from the ALD-Al_2_O_3_ insulator due to the second-annealing treatment. This has been discussed in our previous published report [16]. The ALD-Al_2_O_3_ grown at low temperatures contains a large number of Al-OH bonds. Under the gate voltage electric field, hydrogen atoms of the Al-OH bond would detach and combine with O^2−^ on the surface of a-IGZO to form OH^−^, releasing an electron into the conduction band of a-IGZO, resulting in a negative transfer shift [27]. The second-annealing treatment could reduce the content of Al-OH bonds in Al_2_O_3_ [16], thereby preventing the negative shift in the transfer curve. O-defects would be ionized by the gate bias electric field, and ionized O-defects would release electrons into the conduction band, resulting in a negative shift in the transfer curve. The deviations in *V*_th_ (Δ*V*_th_) between the initial and the 7200 s positive-biased transfer curves at *V*_D_ = 0.1 V were −0.11 V (200 °C), −0.035 V (250 °C), and −0.032 V (300 °C), respectively. This result showed that the PBS stabilities of TG-TFTs had an obvious improvement with increasing first-annealing temperatures, with the TG-TFT first-annealed at 300 °C exhibiting the best PBS stability. The improvement in PBS stabilities might be due to the reduction in *V*_O_ and the increase in bonds between the O atom and metal atom. Compared with In-O and Zn-O, the Ga-O bond energy is larger and more capable of resisting the damage of the gate bias electric field [30].

The NBS results of a-IGZO TG-TFTs were presented in Figure 4. The transfer characteristics of all TG-TFTs exhibited negative shifts after a 7200 s stress duration under the *V*_stress_ = −3 V condition, which is similar to the trends that appeared in other n-type oxide semiconductor (e.g., TiO_2_, SnO_2_) TFTs [31,32], as shown in Figure 4a–c. The negative shifts in *V*_th_ are attributed to the fact that holes were difficult to be generated in n-type oxide semiconductors [16]. The Δ*V*_th_ values between the initial and the 7200 s negative-biased transfer curves at *V*_D_ = 0.1 V were −0.573 V (200 °C), −0.335 V (250 °C), and −0.044 V (300 °C). It was evident that the NBS stabilities of the TG-TFTs were improved with increasing first-annealing temperature, with the 300 °C first-annealed device exhibiting the best NBS stability. The excellent NBS stability would make a-IGZO TG-TFTs very advantageous for applications in the field of transparent displays.

The best PBS and NBS stabilities were both obtained from TG-TFTs with the first-annealing treatment at 300 °C. In order to further understand the effect of the first-annealing treatment on the bias stress stability improvement of TG-TFTs, PL and XPS spectra were conducted on a-IGZO films with first-annealing treatments at 200 °C, 250 °C, and 300 °C.

### 3.3. First-Annealing Effect on a-IGZO Films

Figure 5 and the insert figure showed the PL spectra and separated PL spectra of a-IGZO films with first-annealing at fresh (no first-annealing), 200 °C, 250 °C, and 300 °C. Four broad emission peaks were found in the visible light range of 1.6~3.2 eV, which is consistent with previous reports [33,34,35,36]. From the insert of Figure 5, every peak was observed with a flat top, of which the maximum energy for all peaks was at 2.16 eV. The result meant that the deepest defect in all a-IGZO films was located at 2.16 eV below the conduction band bottom (VBM) [37]. In addition, the energy bands occupied by the flat top of the measured peaks were 0.159 eV (fresh), 0.130 eV (200 °C), 0.107 eV (250 °C), and 0.136 eV (300 °C), respectively. The peak intensity and width of the energy band of the flat top in the PL spectra may illustrate the density and the distribution range of defects in the a-IGZO film [34,35,38]. The flat band width of the a-IGZO film annealed at 300 °C in the PL spectrum decreased by 0.023 eV compared to the fresh a-IGZO film. The bandgap of a-IGZO increased by 0.02 eV after the 300 °C annealing treatment, as shown in the insert of Figure 1b. These results indicated that tail states at the bottom of the conduction band of the a-IGZO film shifted upward by about 0.02 eV after the 300 °C annealing treatment. The PL curve of the a-IGZO film with first-annealing at 250 °C exhibited the highest peak intensity and the narrowest energy band (0.107 eV) of the flat top, while the PL spectra of the a-IGZO film without first-annealing showed the lowest peak intensity and the widest energy band (0.159 eV). These results demonstrated that the O defects in the a-IGZO film with first-annealing at 250 °C were at the deepest energy level, which might result in the largest positive *V*_th_ observed in the corresponding TG-TFT, as shown in Figure 2a,b. The PL spectra of the a-IGZO film with the 300 °C first-annealing treatment showed a lower peak intensity but a wider energy band (0.136 eV) of the flat top compared to the 250 °C first-annealed a-IGZO film. This result might suggest that some shallow defect states were generated in the 300 °C first-annealed a-IGZO film as the overall number of defects decreased [9]. It is known that H_2_O is difficult to be adsorbed on the surface of a-IGZO when the temperature is above 300 °C. In addition, M-OH in a-IGZO reacted with another M-OH to evaporate a H_2_O molecule, and left an ionized oxygen vacancy and two electrons. This is why shallow defect states were generated in the 300 °C first-annealed a-IGZO film [17]. The band diagram explained the changes in the defects observed in the PL spectrum, as shown in Figure 6. In the a-IGZO film first-annealed at 200 °C, the *V*_O_ concentration was significantly reduced compared to the fresh a-IGZO film, as shown in Figure 6b. The a-IGZO film first-annealed at 250 °C has a higher *V*_O_ concentration than those first-annealed at 200 °C and 300 °C, with the *V*_O_ states in a-IGZO first-annealed at 250 °C occupying the central region of the bandgap, as shown in Figure 6c. In contrast, the a-IGZO film first-annealed at 300 °C had a lower *V*_O_ concentration compared to that first-annealed at 200 °C, but the *V*_O_ were mainly distributed at shallower energy levels below the conduction band, as illustrated in Figure 6d. And this might be the reason for the higher mobility observed in the TG-TFT with the 300 °C first-annealing treatment compared to the TG-TFT with 250 °C first-annealed a-IGZO, as shown in Table 1.

It is reported that the bonding states of gallium atoms in a-IGZO had a great impact on *V*_th_, *SS*, and *μ* of a-IGZO TFTs [39,40]. It is known that the Ga-O bond exhibits the shortest bond length and the strongest binding energy compared to the In-O bond and Zn-O bond [30]. The XPS spectra of oxygen and gallium elements in a-IGZO films with first-annealing treatments at 200–300 °C were employed to investigate the relationship between transfer characteristics of TG-TFTs and the first-annealing temperature. The first-annealing temperature of the controlled samples were set at fresh (no first-annealing treatment), 200 °C, 250 °C, and 300 °C, respectively. Figure 7a,b showed the O 1s and Ga 3d XPS spectra of a-IGZO films with first-annealing treatments at fresh, 200 °C, 250 °C, and 300 °C, respectively. The O 1s spectra could be deconvoluted into two sub-peaks based on Gaussian fitting; i.e., the bond between O atom and metal atom (O-M) (centered around 530.15 eV) and *V*_O_ (centered around 531.47 eV). The Ga 3d spectra could be deconvoluted into Ga-Ga (centered around 17.8 eV) and Ga-O (centered around 19.60 eV) [30,41,42,43]. The O-M core peaks of a-IGZO films with first-annealing at fresh, 200 °C, 250 °C, and 300 °C were at 530.02 eV, 530.12 eV, 530.20 eV, and 530.15 eV with an error of ±0.02 eV, while *V*_O_ core peaks were at 530.85 eV, 531.45 eV, 531.55 eV, and 531.50 eV with an error of ±0.02 eV, respectively, as shown in Figure 6a. The core peak energy values of O-M for the a-IGZO films were almost consistent. The *V*_O_ core peak energy for the a-IGZO film with first-annealing at 250 °C was found to be the biggest (531.55 eV), which indicated that the defect states in the a-IGZO film were located at the deepest energy level compared to that in other a-IGZO films, consistent with the PL spectra results from Figure 5. The contents of *V*_O_ and O-M bonds were calculated from the O 1s spectra, as shown in Table 2. The *V*_O_ contents in a-IGZO films were 26.60% (fresh), 11.17% (200 °C), 12.84% (250 °C), and 11.01% (300 °C), respectively. *V*_O_ at shallow energy levels act as shallow donors, releasing electrons into the conduction band under the gate voltage; *V*_O_ at deep energy levels would be ionized and capture electrons under the gate voltage field, leading to a decrease in mobility. For the IGZO film first-annealed at 250 °C, the *V*_O_ content was higher than that in a-IGZO films first-annealed at 200 °C and 300 °C, as shown in Table 2. Additionally, PL spectra showed that the number of deep defects in the a-IGZO film first-annealed at 250 °C was greater than that in a-IGZO films first-annealed at 200 °C and 300 °C, as shown in Figure 5. Based on these two results, it suggested that the *V*_O_ in the a-IGZO film first-annealed at 250 °C might occupy the central region of the bandgap, as shown in Figure 6c. The mobility of the 250 °C first-annealed TG-TFT decreased after a significant number of electrons were captured by the defects at the center of the bandgap under the gate voltage as shown in Table 1. On the other hand, the a-IGZO film first-annealed at 300 °C had a lower *V*_O_ concentration compared to the a-IGZO film first-annealed at 200 °C, but these vacancies were mainly distributed at shallower energy levels below the conduction band. As shown in Figure 6d, the *V*_O_ states occupying these shallow energy levels near the bottom of the conduction band released electrons into the conduction band under the gate voltage field, resulting in an increase in the carrier concentration and mobility. The contents of O-M bonds in a-IGZO films were 17.27% (fresh), 32.57% (200 °C), 30.45% (250 °C), and 32.44% (300 °C), respectively. The first-annealing treatment led to a 13.76–15.59% decrease in *V*_O_ content and a 13.18–15.3% increase in the O-M bond content for a-IGZO films, as shown in Table 2. The change in the contents of *V*_O_ and O-M bonds demonstrated that the O-related defects in a-IGZO films decreased with the first-annealing treatment temperature. The *V*_O_ content in the 250 °C first-annealed a-IGZO film was slightly higher than that in a-IGZO films with first-annealing at 200 °C and 300 °C, indicating that the combination state between the O and metal atoms under this condition was different compared to other a-IGZO films.

In this report, the Ga-Ga peaks in all a-IGZO films were centered at 17.80 eV with a calculation error of ±0.02 eV. The Ga-O peak in the 250 °C first-annealed a-IGZO film was centered at 19.75 eV (Red star in Figure 7b), whereas the Ga-O peaks for other a-IGZO films were all centered at 19.60 eV with a calculation error of ±0.02 eV, as shown in Figure 7b. The 0.15 eV increase in the Ga-O peak indicated that the coordination between Ga and O in the 250 °C first-annealed a-IGZO film was better than that in other a-IGZO films. The Ga-O contents in a-IGZO films were 12.82% (fresh), 12.92% (200 °C), 11.18% (250 °C), and 13.34% (300 °C), respectively. The a-IGZO film with the first-annealing temperature at 300 °C had the highest Ga-O content. This might be the reason why TG TFTs with first-annealing at 300 °C exhibited the best PBS and NBS stability as shown in Figure 3c and Figure 4c, based on the fact that Ga-O could suppress the ionization of *V*_O_ under gate bias [30].

Usually, an increase in *V*_O_ in a-IGZO would lead to the *V*_th_ shifting in a negative direction, while the enhancement in the Ga-O bond would lead to the *V*_th_ shifting to positively. In comparison to the a-IGZO film with the first-annealing temperature at 200 °C, the 300 °C first-annealed a-IGZO film showed a 0.16% decrease in *V*_O_ content and a 0.42% increase in Ga-O content. As a result, the *V*_th_ of the TG-TFT with the 300 °C first-annealed a-IGZO exhibited a positive shift. The *V*_O_ content in the 250 °C first-annealed a-IGZO film was higher than that in the 200 °C and 300 °C annealed films, and while the Ga-O content in the 250 °C first-annealed a-IGZO film was lower than that in the 200 °C and 300 °C annealed films. Thus, the *V*_th_ of the TG-TFT with the 250 °C first-annealed a-IGZO should be more negative than that for the TG-TFTs with other a-IGZO films. However, the largest positive *V*_th_ was observed at the TG-TFT with the 250 °C first-annealed a-IGZO. This strange phenomenon might suggest that the Ga-O content had a greater impact on the *V*_th_ than *V*_O_ content. And the better coordination between Ga and O promoted the idea that the *V*_th_ of the TG-TFT with the 250 °C first-annealed a-IGZO film was more positive than that for the 300 °C first-annealed TG-TFT.

## 4. Conclusions

In this work, the electrical properties and stabilities of a-IGZO TG-TFTs were systematically investigated by controlling the first-annealing temperature, and the mechanism of the first-annealing treatment on a-IGZO films was explored through PL spectra and XPS analysis. The TG-TFTs achieved good switching function when the first-annealing temperature at 200 °C–300 °C. The TG-TFT with the first-annealing treatment at 300 °C demonstrated the best electrical properties, PBS stability, and NBS stability.

The PL spectra and XPS spectra revealed the change in the O defect states and Ga-O bond state in the first-annealed a-IGZO films. O defects were reduced with first-annealing treatments, but some shallow states were produced when the first-annealing temperature increased to 300 °C. The best coordination situation between Ga and O atoms occurred for the a-IGZO film with the first-annealing treatment at 250 °C, leading to the biggest positive *V*_th_ of TG-TFTs. Furthermore, the highest content of the Ga-O bond and minimal defect intensity were both obtained in the a-IGZO film with first-annealing at 300 °C, which contributed to the excellent PBS and NBS stabilities of TG-TFTs. The interplay between O defects reduction and Ga-O bond strengthening underscores the importance of annealing temperature in balancing defect passivation and carrier transport for high-performance, transparent flexible electronics. This work provides a strategic framework for tailoring annealing processes to achieve highly stable a-IGZO TG-TFTs, advancing their applicability in next-generation transparent and wearable technologies.

## Figures and Tables

**Figure 1 nanomaterials-15-00460-f001:**
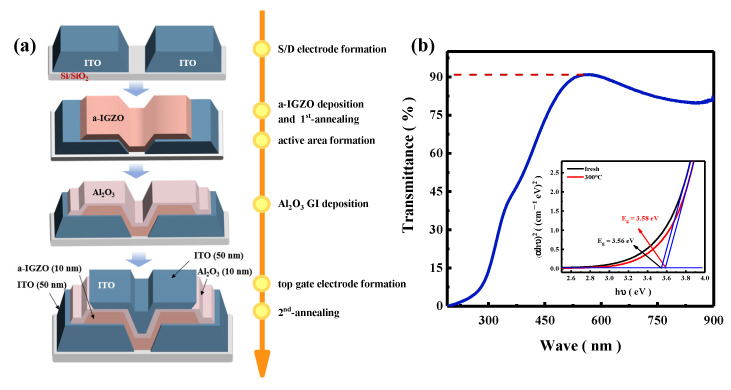
(**a**) Schematic of fabrication process, and (**b**) the measured transmittance of the a-IGZO TG-TFT device across various wavelengths, and the insert is the calculated band gap energy for a 130 nm a-IGZO film on a quartz substrate.

**Figure 2 nanomaterials-15-00460-f002:**
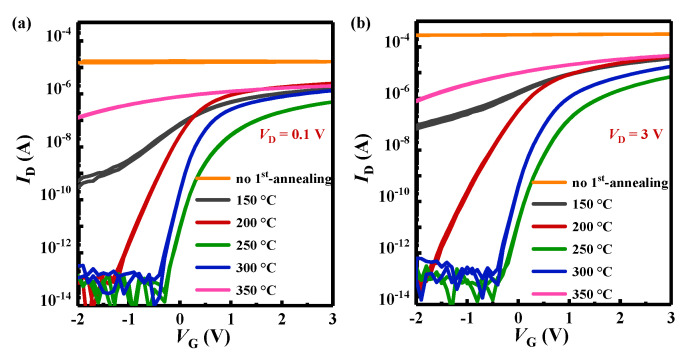
(**a**,**b**) These are transfer characteristics of TG-TFTs with different first-annealing treatments under *V*_D_ = 0.1 V and *V*_D_ = 3 V, respectively.

**Figure 3 nanomaterials-15-00460-f003:**
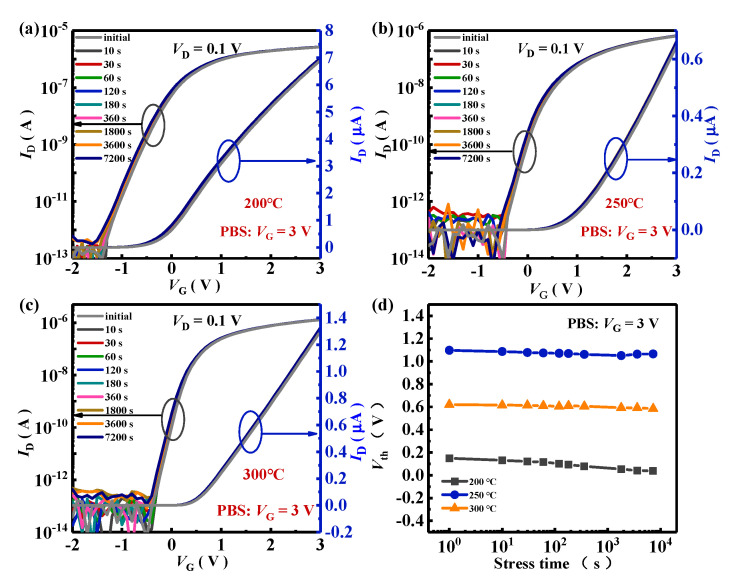
PBS time evolution (under *V*_stress_ = 3 V and *V*_D_ = 0.1 V at RT without illumination) of the transfer for a-IGZO TG-TFTs with first-annealing treatments at (**a**) 200 °C, (**b**) 250 °C, and (**c**) 300 °C, and (**d**) *V*_th_ shifts during the PBS process.

**Figure 4 nanomaterials-15-00460-f004:**
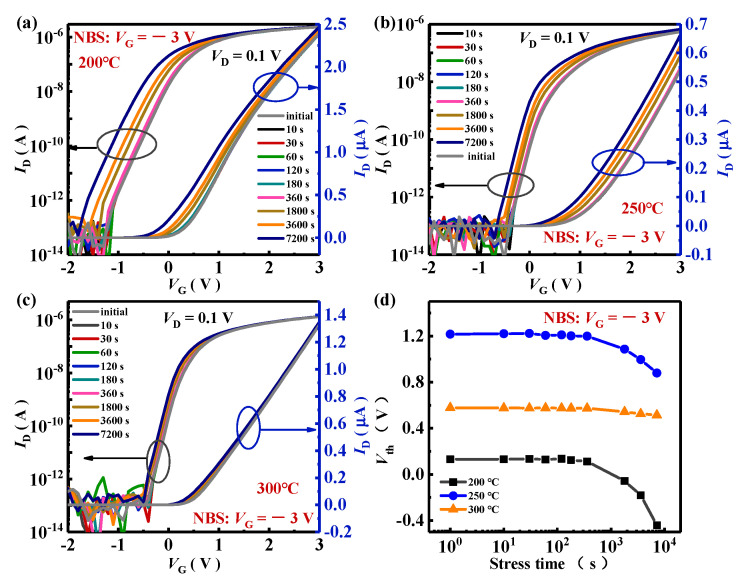
NBS time evolution (under *V*_stress_ = −3 V and *V*_D_ = 0.1 V at RT without illumination) of the transfer for a-IGZO TG-TFTs with first-annealing treatments at (**a**) 200 °C, (**b**) 250 °C, and (**c**) 300 °C, and (**d**) *V*_th_ shifts during the NBS process.

**Figure 5 nanomaterials-15-00460-f005:**
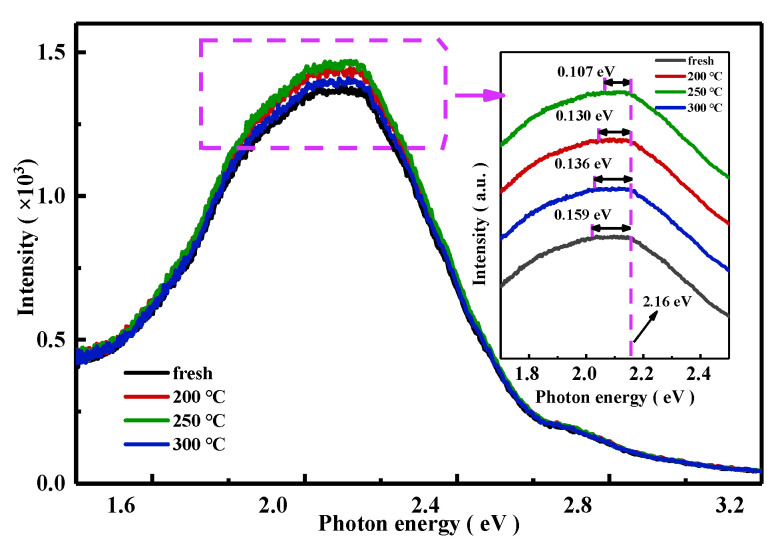
PL spectra for a-IGZO films under different first-annealing temperatures of fresh, 200 °C, 250 °C, and 300 °C, respectively; insert is separated PL spectra in the range of 1.7–2.5 eV.

**Figure 6 nanomaterials-15-00460-f006:**
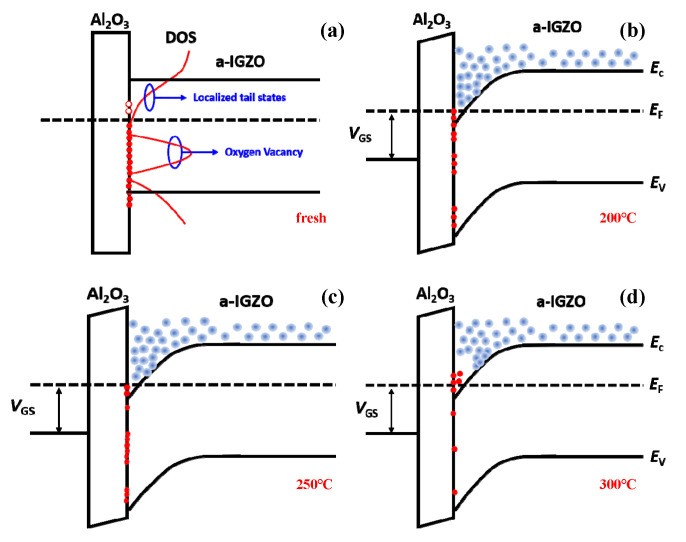
Energy bands and carrier transport diagrams of TG-TFTs with different first-annealing temperatures of fresh, 200 °C, 250 °C, and 300 °C: (**a**) subthreshold region for the fresh TG-TFT, and on-state regions for (**b**) 200 °C first-annealed TG-TFT, (**c**) 250 °C first-annealed TG-TFT, and (**d**) 300 °C first-annealed TG-TFT, respectively.

**Figure 7 nanomaterials-15-00460-f007:**
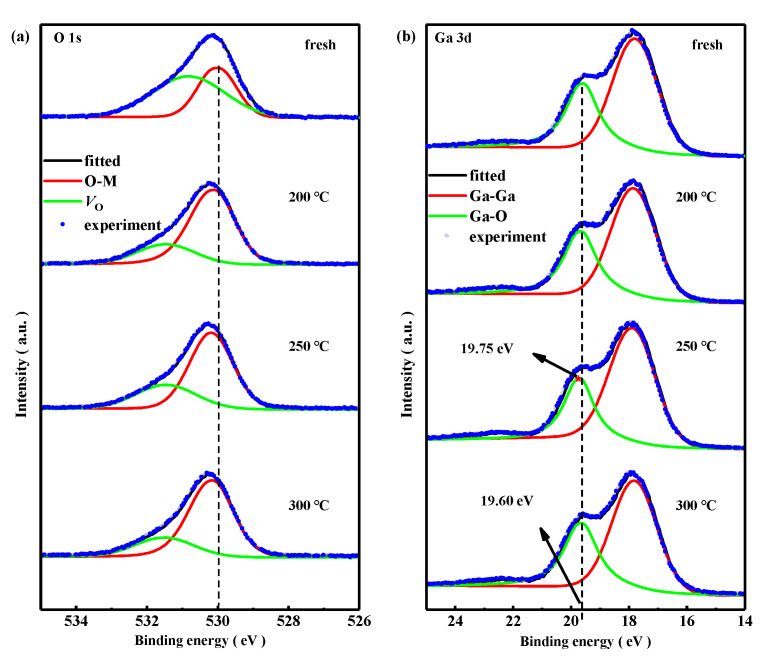
(**a**) Deconvoluted O 1s XPS profile with *V*_O_ and O-M, and (**b**) deconvoluted Ga 3d XPS profile with Ga-Ga and Ga-O in a-IGZO films received different first-annealing treatments at fresh, 200 °C, 250 °C, and 300 °C, respectively.

**Table 1 nanomaterials-15-00460-t001:** Electrical characteristic parameters of TG-TFTs with different first-annealing temperature.

	Parameters	*V*_th-lin_ (V)	*V*_th-sat_ (V)	*μ*_sat_ (cm^2^ V^−1^ s^−1^)	*I*_on_/*I*_off_ (*V*_D_ = 3 V)	*SS* (mV/dec)	*N*_t_ (eV^−1^ cm^−2^)
T (°C)							
150		−1.13	−1.23	9.13	4.03 × 10^3^	--	--
200		−0.53	−0.83	30.52	4.46 × 10^8^	203	6.10 × 10^12^
250		1.12	0.78	6.87	6.98 × 10^7^	137	3.28 × 10^12^
300		0.52	0.33	13.05	1.73 × 10^8^	130	2.98 × 10^12^
350		−2.05	−3.05	--	7.2	--	--

**Table 2 nanomaterials-15-00460-t002:** Bond content percentage of IGZO films with different first-annealing treatments.

Annealing Condition (°C)	Content (%)			
	*V* _O_	O-M	Ga-Ga	Ga-O
fresh	26.60	17.27	23.74	12.82
200	11.17	32.57	23.81	12.92
250	12.84	30.45	25.85	11.18
300	11.01	32.44	23.6	13.34

## Data Availability

The data are contained within the article.

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
