# Peer review of "Enhancement in Performance and Reliability of Fully Transparent a-IGZO Top-Gate Thin-Film Transistors by a Two-Step Annealing Treatment"

_nanomaterials, 2025, doi:10.3390/nano15060460_

Round 1
Reviewer 1 Report
Comments and Suggestions for Authors
This study suggests two step annealing process for fabrication IGZO TFTs. Interestingly, significant electrical properties by controlling 1st annealing temperature.
1 1st annealing was conducted right after deposition of IGZO. Is there any change of material structure after 1st annealing?
2 Figure 1a shows the device fabrication process. In the figure, there is wide overlap area between the gate and the source/drain electrode. There may be serious parasitic capacitance. Please discuss about this.
3 The author explained that the significant shfit of Vth with NBS is originated from H-atom doping during ALD deposition and 2nd annealing. There might be some diffusion of hydrogen atom into IGZO film. Is there any proper data to show this?
4 Figure 5 shows PL data with various annealing temperature. The study suggests deep states in the band gap by PL data. Please add band diagram with band gap and deep energy states for readers to understand batter.
Author Response
Dear Editor and reviewers:
Thank you very much for your advice. We appreciate the constructive and useful suggestions for the improvement of our manuscript (nanomaterials-3479640). The manuscript has been revised according to the comments. Detailed responses to individual comments were listed as the attachment.
Best Regards!
Yuxiang Li
(On behalf of all co-authors)

Reviewer 2 Report
Comments and Suggestions for Authors
Transparency and Optical Properties
- The optical bandgap of a-IGZO was calculated to be 3.84 eV, which is slightly higher than typical values (3.0–3.6 eV).
→ Is the increase in bandgap related to the reduction of oxygen defects due to the 1st-annealing treatment?
→ How can the relationship between bandgap increase and changes in trap states be further examined using PL analysis? - PL spectra showed that defect density decreased with the 1st-annealing treatment, but shallow defect states were generated at 300°C.
→ Should PL analysis be further refined to distinguish defect characteristics at 250°C and 300°C more precisely?
→ Can Raman spectroscopy or EPR (Electron Paramagnetic Resonance) be used for more quantitative analysis of oxygen defects?
Electrical Characteristics and Switching Behavior
- TG-TFTs without the 1st-annealing treatment showed no switching characteristics.
→ How exactly does the 1st-annealing treatment affect the reduction of oxygen defects and interfacial states during Alâ‚‚O₃ deposition?
→ Can XPS analysis be further used to examine chemical bonding changes at the Alâ‚‚O₃/a-IGZO interface? - Only devices annealed at 200°C–300°C exhibited good switching characteristics, while those at 150°C and 350°C did not.
→ At 350°C, the deterioration of switching behavior was attributed to the formation of new trap states. Is this primarily due to an increase in oxygen defects, or could structural changes in a-IGZO also be a factor?
→ Should additional structural analysis, such as XRD (X-ray Diffraction) or TEM (Transmission Electron Microscopy), be conducted to confirm structural variations? - Although the best switching characteristics were expected at 250°C, the highest mobility (μ) was observed at 300°C.
→ Why does the presence of shallow donor states at 300°C still result in higher mobility?
→ Should Hall Effect Measurement or C-V analysis be conducted to further investigate carrier concentration and mobility variations?
Bias Stress and Device Reliability
- PBS and NBS stability improved with increasing 1st-annealing temperature.
→ The ΔVth shift decreased from -0.11V (200°C) → -0.035V (250°C) → -0.032V (300°C). What is the precise mechanism behind this improvement?
→ How do changes in oxygen defects (O-defects) and Ga-O bonding contribute to the stabilization of ΔVth? - H-doping effects were introduced by ALD-Alâ‚‚O₃ deposition, and 2nd-annealing treatment partially removed hydrogen at the interface.
→ How does hydrogen incorporation and removal at the interface affect the long-term electrical properties of the device?
→ Should SIMS (Secondary Ion Mass Spectrometry) analysis be performed to quantitatively measure hydrogen concentration at the interface? - NBS results showed a significant reduction in ΔVth with increasing 1st-annealing temperature (-0.573V → -0.335V → -0.044V).
→ What is the direct correlation between increased Ga-O bonding and improved ΔVth stability?
→ Do similar trends appear in other n-type oxide semiconductors (e.g., TiOâ‚‚, SnOâ‚‚), or is this unique to a-IGZO?
Ga-O Bonding and Oxygen Defect Analysis
- XPS analysis showed that the highest Ga-O bonding content was observed at 300°C.
→ How does the increase in Ga-O bonding contribute to improved electron mobility and device reliability?
→ Should additional experimental validation be conducted to confirm the impact of Ga-O bonding on carrier trapping and mobility? - O 1s XPS spectra showed that VO content increased at 250°C but decreased again at 300°C.
→ Does the increase in VO at 250°C contribute to enhanced mobility, or are there other competing factors?
→ Can the optimal a-IGZO structure be engineered by fine-tuning the VO/Ga-O ratio?
Author Response

(The authors gave the same response as above.)
